# Autowarp: Learning a Warping Distance from Unlabeled Time Series Using Sequence Autoencoders

**Abubakar Abid**
Stanford University
a12d@stanford.edu

**James Zou**
Stanford University
jamesz@stanford.edu

## Abstract

Measuring similarities between unlabeled time series trajectories is an important problem in domains as diverse as medicine, astronomy, finance, and computer vision. It is often unclear what is the appropriate metric to use because of the complex nature of noise in the trajectories (e.g. different sampling rates or outliers). Domain experts typically hand-craft or manually select a specific metric, such as dynamic time warping (DTW), to apply on their data. In this paper, we propose Autowarp, an end-to-end algorithm that optimizes and learns a good metric given unlabeled trajectories. We define a flexible and differentiable family of warping metrics, which encompasses common metrics such as DTW, Euclidean, and edit distance. Autowarp then leverages the representation power of sequence autoencoders to optimize for a member of this warping distance family. The output is a metric which is easy to interpret and can be robustly learned from relatively few trajectories. In systematic experiments across different domains, we show that Autowarp often outperforms hand-crafted trajectory similarity metrics.

## 1 Introduction

A *time series*, also known as a *trajectory*, is a sequence of observed data $\boldsymbol{t} = (t_1, t_2, ...t_n)$ measured over time. A large number of real world data in medicine [1], finance [2], astronomy [3] and computer vision [4] are time series. A key question that is often asked about time series data is: "How similar are two given trajectories?" A notion of trajectory similarity allows one to do unsupervised learning, such as clustering and visualization, of time series data, as well as supervised learning, such as classification [5]. However, measuring the distance between trajectories is complex, because of the temporal correlation between data in a time series and the complex nature of the noise that may be present in the data (e.g. different sampling rates) [6].

In the literature, many methods have been proposed to measure the similarity between trajectories. In the simplest case, when trajectories are all sampled at the same frequency and are of equal length, Euclidean distance can be used [7]. When comparing trajectories with different sampling rates, dynamic time warping (DTW) is a popular choice [7]. Because the choice of distance metric can have a significant effect on downstream analysis [5, 6, 8], a plethora of other distances have been hand-crafted based on the specific characteristics of the data and noise present in the time series.

However, a review of five of the most popular trajectory distances found that no one trajectory distance is more robust than the others to all of the different kinds of noise that are commonly present in time series data [8]. As a result, it is perhaps not surprising that many distances have been manually designed for different time series domains and datasets. In this work, we propose an alternative to hand-crafting a distance: we develop an end-to-end framework to *learn* a good similarity metric directly from unlabeled time series data.

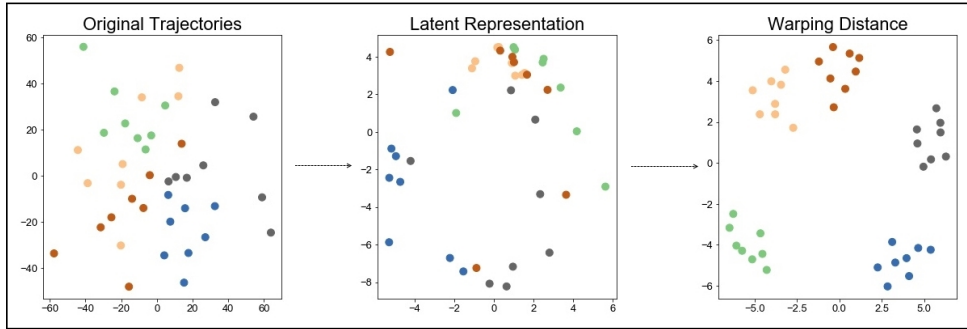

Figure 1: **Learning a Distance with Autowarp.** Here we visualize the stages of Autowarp by using multi-dimensional scaling (MDS) to embed a set of 50 trajectories into two dimensions at each step of the algorithm. Each dot represents one observed trajectory that is generated by adding Gaussian noise and outliers to 10 copies of 5 seed trajectories (each color represents a seed). (left) First, we run MDS on the original trajectories with Euclidean distance. (center) Next, we run MDS on the latent representations learned with a sequence-to-sequence autoencoder, which partially resolves the original clusters. (right) Finally, we run MDS on the original trajectories using the learned warping distance, which completely resolves the original clusters.

While data-dependent analysis of time-series is commonly performed in the context of supervised learning (e.g. using RNNs or convolutional networks to classify trajectories [9]), this is not often performed in the case when the time series are unlabeled, as it is more challenging to determine notions of similarity in the absence of labels. Yet the unsupervised regime is critical, because in many time series datasets, ground-truth labels are difficult to determine, and yet the notion of similarity plays a key role. For example, consider a set of disease trajectories recorded in a large electronic health records database: we have the time series information of the diseases contracted by a patient, and it may be important to determine which patient in our dataset is most similar to another patient based on his or her disease trajectory. Yet, the choice of ground-truth labels is ambiguous in this case. In this work, we develop an easy-to-use method to determine a distance that is appropriate for a given set of unlabeled trajectories.

In this paper, we restrict ourselves to the family of trajectory distances known as *warping distances* (formally defined in Section 2). This is for several reasons: warping distances have been widely studied, and are intuitive and interpretable [7]; they are also efficient to compute, and numerous heuristics have been developed to allow nearest-neighbor queries on datasets with as many as trillions of trajectories [10]. Thirdly, although they are a flexible and general class, warping distances are particularly well-suited to trajectories, and serve as a means of regularizing the unsupervised learning of similarity metrics directly from trajectory data. We show through systematic experiments that learning an appropriate warping distance can provide insight into the nature of the time series data, and can be used to cluster, query, or visualize the data effectively.

**Related Work**   The development of distance metrics for time series stretches at least as far back as the introduction of dynamic time warping (DTW) for speech recognition [11]. Limitations of DTW led to the development and adoption of the Edit Distance on Real Sequence (EDR) [12], the Edit Distance with Real Penalty (ERP) [13], and the Longest Common Subsequence (LCSS) [14] as alternative distances. Many variants of these distances have been proposed, based on characteristics specific to certain domains and datasets, such as the Symmetric Segment-Path Distance (SSPD) [7] for GPS trajectories, Subsequence Matching [15] for medical time series data, among others [16].

Prior work in metric learning from trajectories is generally limited to the supervised regime. For example, in recent years, convolutional neural networks [9], recurrent neural networks (RNNs) [17], and siamese recurrent neural networks [18] have been proposed to classify neural networks based on labeled training sets. There has also been some work in applying unsupervised deep learning learning to time series [19]. For example, the authors of [20] use a pre-trained RNN to extract features from time-series that are useful for downstream classification. Unsupervised RNNs have also found use in anomaly detection [21] and forecasting [22] of time series.

## 2 The Autowarp Approach

Our approach, which we call Autowarp, consists of two steps. First, we learn a latent representation for each trajectory using a sequence-to-sequence autoencoder. This representation takes advantage of the temporal correlations present in time series data to learn a low-dimensional representation of each trajectory. In the second stage, we search a family of warping distances to identify the warping distance that, when applied to the original trajectories, is most similar to the Euclidean distances between the latent representations. Fig. 1 shows the application of Autowarp to synthetic data.

**Learning a latent representation** Autowarp first learns a latent representation that captures the significant properties of trajectories in an unsupervised manner. In many domains, an effective latent representation can be learned by using autoencoders that reconstruct the input data from a low-dimensional representation. We use the same approach using sequence-to-sequence autoencoders.

This approach is inspired by similar sequence-to-sequence autoencoders, which have been successfully applied to sentiment classification [23], machine translation [24], and learning representations of videos [25]. In the architecture that we use (illustrated in Fig. 2), we feed each step in the trajectory sequentially into an *encoding* LSTM layer. The hidden state of the final LSTM cell is then fed identically into a *decoding* LSTM layer, which contains as many cells as the length of the original trajectory. This layer attempts to reconstruct each trajectory based solely on the learned latent representation for that trajectory.

What kind of features are learned in the latent representation? Generally, the hidden representation captures overarching features of the trajectory, while learning to ignore outliers and sampling rate. We illustrate this in Fig. S1 in Appendix A: the LSTM autoencoders learn to denoise representations of trajectories that have been sampled at different rates, or in which outliers have been introduced.

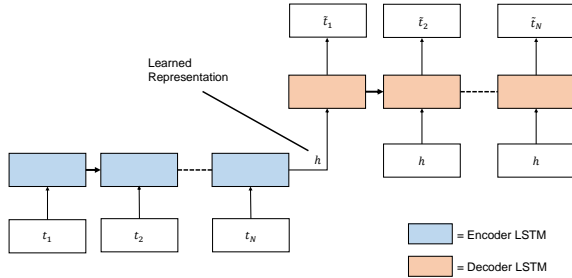

Figure 2: **Schematic for LSTM Sequence-Sequence Autoencoder.** We learn a latent representation for each trajectory by passing it through a sequence-to-sequence autoencoder that is trained to minimize the reconstruction loss $\left\| t - \tilde{t} \right\|^2$ between the original trajectory $t$ and decoded trajectory $\tilde{t}$. In the decoding stage, the latent representation $h$ is passed as input into each LSTM cell.

**Warping distances** Once a latent representation is learned, we search from a family of warping distances to find the warping distance across the original trajectories that mimics the distances between each trajectory's latent representations. This can be seen as "distilling" the representation learned by the neural network into a warping distance (e.g. see [26]). In addition, as warping distances are generally well-suited to trajectories, this serves to regularize the process of distance metric learning, and generally produces better distances than using the latent representations directly (as illustrated in Fig. 1).

We proceed to formally define a *warping distance*, as well as the family of warping distances that we work with for the rest of the paper. First, we define a *warping path* between two trajectories.

**Definition 1.** *A **warping path** $p = (p_0, \ldots p_L)$ between two trajectories $t^A = (t_1^A, \ldots t_n^A)$ and $t^B = (t_1^B, \ldots t_m^B)$ is a sequence of pairs of trajectory states, where the first state comes from trajectory $t^A$ or is null (which we will denote as $t_0^A$), and the second state comes from trajectory $t^B$ or is null (which we will denote as $t_0^B$). Furthermore, $p$ must satisfy two properties:*

- *boundary conditions: $p_0 = (t_0^A, t_0^B)$ and $p_L = (t_n^A, t_m^B)$*

- *valid steps: $p_k = (t_i^A, t_j^B) \implies p_{k+1} \in \{(t_{i+1}^A, t_j^B), (t_{i+1}^A, t_{j+1}^B), (t_i^A, t_{j+1}^B)\}$.*

Warping paths can be seen as traversals on a $(n+1)$-by-$(m+1)$ grid from the bottom left to the top right, where one is allowed to go up one step, right one step, or one step up and right, as shown in Fig. S2 in Appendix A. We shall refer to these as *vertical*, *horizontal*, and *diagonal* steps respectively.

**Definition 2.** *Given a set of trajectories $\mathcal{T}$, a **warping distance** $d$ is a function that maps each pair of trajectories in $\mathcal{T}$ to a real number $\in [0, \infty)$. A warping distance is completely specified in terms of a cost function $c(\cdot, \cdot)$ on two pairs of trajectory states:*

*Let $\boldsymbol{t}^A, \boldsymbol{t}^B \in \mathcal{T}$. Then $d(\boldsymbol{t}^A, \boldsymbol{t}^B)$ is defined[1] as $d(\boldsymbol{t}^A, \boldsymbol{t}^B) = \min_{\boldsymbol{p} \in P} \sum_{i=1}^{L} c(p_{i-1}, p_i)$*

*The function $c(p_{i-1}, p_i)$ represents the cost of taking the step from $p_{i-1}$ to $p_i$, and, in general, differs for horizontal, vertical, and diagonal steps. $P$ is the set of all warping paths between $\boldsymbol{t}^A$ and $\boldsymbol{t}^B$.*

Thus, a warping distance represents a particular optimization carried over all valid warping paths between two trajectories. In this paper, we define a family of warping distances $\mathcal{D}$, with the following parametrization of $c(\cdot, \cdot)$:

$$c((t_i^A, t_j^B), (t_{i'}^A, t_{j'}^B)) = \begin{cases} \sigma(\|t_{i'}^A - t_{j'}^B\|, \frac{\epsilon}{1-\epsilon}) & i' > i, j' > j \\ \frac{\alpha}{1-\alpha} \cdot \sigma(\|t_{i'}^A - t_{j'}^B\|, \frac{\epsilon}{1-\epsilon}) + \gamma & i' = i \text{ or } j' = j \end{cases} \quad (1)$$

Here, we define $\sigma(x, y) \overset{\text{def}}{=} y \cdot \tanh(x/y)$ to be a soft thresholding function, such that $\sigma(x, y) \approx x$ if $0 \leq x \leq y$ and $\sigma(x, y) \approx y$ if $x > y$. And, $\sigma(x, \infty) \overset{\text{def}}{=} x$. The family of distances $\mathcal{D}$ is parametrized by three parameters $\alpha, \gamma, \epsilon$. With this parametrization, $\mathcal{D}$ includes several commonly used warping distances for trajectories, as shown in Table 1, as well as many other warping distances.

| Trajectory Distance | $\alpha$ | $\gamma$ | $\epsilon$ |
|---|---|---|---|
| Euclidean[a] | 1 | 0 | 1 |
| Dynamic Time Warping (DTW) [11] | 0.5 | 0 | 1 |
| Edit Distance ($\gamma_0$) [13] | 0 | $0 < \gamma_0$ | 1 |
| Edit Distance on Real Sequences ($\gamma_0, \epsilon_0$) [12] [b] | 0 | $0 < \gamma_0$ | $0 < \epsilon_0 < 1$ |

Table 1: Parametrization of common trajectory dissimilarities

[a]The Euclidean distance between two trajectories is infinite if they are of different lengths
[b]This is actually a smooth, differentiable approximation to EDR

**Optimizing warping distance using betaCV**    Within our family of warping distances, how do we choose the one that aligns most closely with the learned latent representation? To allow a comparison between latent representations and trajectory distances, we use the concept of betaCV:

**Definition 3.** *Given a set of trajectories $\mathcal{T} = \{\boldsymbol{t}^1, \boldsymbol{t}^2, \dots \boldsymbol{t}^T\}$, a trajectory metric $d$ and an assignment to clusters $C(i)$ for each trajectory $\boldsymbol{t}^i$, the **betaCV**, denoted as $\beta$, is defined as:*

$$\beta(d) = \frac{\frac{1}{Z} \sum_{i=1}^{T} \sum_{j=1}^{T} d(\boldsymbol{t}^i, \boldsymbol{t}^j) \, \mathbb{1}\left[C(i) = C(j)\right]}{\frac{1}{T^2} \sum_{i=1}^{T} \sum_{j=1}^{T} d(\boldsymbol{t}^i, \boldsymbol{t}^j)}, \quad (2)$$

*where $Z = \sum_{i=1}^{T} \sum_{j=1}^{T} \mathbb{1}\left[C(i) = C(j)\right]$ is the normalization constant needed to transform the numerator into an average of distances.*

In the literature, the betaCV is used to evaluate different clustering assignments $C$ for a fixed distance [27]. In our work, it is the distance $d$ that is not known; were true cluster assignments known, the betaCV would be a natural quantity to minimize over the distances in $\mathcal{D}$, as it would give us a distance metric that minimizes the average distance of trajectories to other trajectories within the same cluster (normalized by the average distances across all pairs of trajectories).

However, as the clustering assignments are not known, we instead use the Euclidean distances between to the latent representations of two trajectories to determine whether they belong to the same "cluster." In particular, we designate two trajectories as belonging to the same cluster if the distance between their latent representations is less than a threshold $\delta$, which is chosen as a percentile $\bar{p}$ of the

distribution of distances between all pairs of latent representations. We will denote this version of the betaCV, calculated based on the latent representations learned by an autoencoder, as $\hat{\beta}_h(d)$:

**Definition 4.** *Given a set of trajectories $\mathcal{T} = \{\boldsymbol{t}^1, \boldsymbol{t}^2, \dots \boldsymbol{t}^T\}$, a metric $d$ and a latent representation for each trajectory $h_i$, the **latent betaCV**, denoted as $\hat{\beta}$, is defined as:*

$$\hat{\beta} = \frac{\frac{1}{Z} \sum_{i=1}^{T} \sum_{j=1}^{T} d(\boldsymbol{t}^i, \boldsymbol{t}^j)\, \mathbb{1}\left[\|h_i - h_j\|_2 < \delta\right]}{\frac{1}{T^2} \sum_{i=1}^{T} \sum_{j=1}^{T} d(\boldsymbol{t}^i, \boldsymbol{t}^j)}, \tag{3}$$

*where $Z$ is a normalization constant defined analogously as in (2). The threshold distance $\delta$ is a hyperparameter for the algorithm, generally set to be a certain threshold percentile ($\bar{p}$) of all pairwise distances between latent representations.*

With this definition in hand, we are ready to specify how we choose a warping distance based on the latent representations. We choose the warping distance that gives us the lowest latent betaCV:

$$\boxed{\hat{d} = \underset{d \in \mathcal{D}}{\arg\min}\, \hat{\beta}(d).}$$

We have seen that the learned representations $h_i$ are not always able to remove the noise present in the observed trajectories. It is natural to ask, then, whether it is a good idea to calculate the betaCV using the noisy latent representations, in place of true clustering assignments. In other word, suppose we computed $\beta$ based on known clusters assignments in a trajectory dataset. If we then computed $\hat{\beta}$ based on somewhat noisy learned latent representations, could it be that $\beta$ and $\hat{\beta}$ differ markedly? In Appendix C, we carry out a theoretical analysis, assuming that the computation of $\hat{\beta}$ is based on a noisy clustering $\tilde{C}$. We present the conclusion of that analysis here:

**Proposition 1** (**Robustness of Latent BetaCV**). *Let $d$ be a trajectory distance defined over a set of trajectories $\mathcal{T}$ of cardinality $T$. Let $\beta(d)$ be the betaCV computed on the set of trajectories using the true cluster labels $\{C(i)\}$. Let $\hat{\beta}(d)$ be the betaCV computed on the set of trajectories using noisy cluster labels $\{\tilde{C}(i)\}$, which are generated by independently randomly reassigning each $C(i)$ with probability $p$. For a constant $K$ that depends on the distribution of the trajectories, the probability that the latent betaCV changes by more than $x$ beyond the expected $Kp$ is bounded by:*

$$\Pr(|\beta - \hat{\beta}| > Kp + x) \leq e^{-2Tx^2/K^2} \tag{4}$$

This result suggests that a latent betaCV computed based on latent representations may still be a reliable metric even when the latent representations are somewhat noisy. In practice, we find that the quality of the autoencoder does have an effect on the quality of the learned warping distance, up to a certain extent. We quantify this behavior using an experiment showin in Fig. S4 in Appendix A.

## 3    Efficiently Implementing Autowarp

There are two computational challenges to finding an appropriate warping distance. One is efficiently searching through the continuous space of warping distances. In this section, we show that the computation of the BetaCV over the family of warping distances defined above is differentiable with respect to quantities $\alpha, \gamma, \epsilon$ that parametrize the family of warping distances. Computing gradients over the whole set of trajectories is still computationally expensive for many real-world datasets, so we introduce a method of sampling trajectories that provides significant speed gains. The formal outline of Autowarp is in Appendix B.

**Differentiability of betaCV.**    In Section 2, we proposed that a warping distance can be identified by the distance $d \in \mathcal{D}$ that minimizes the BetaCV computed from the latent representations. Since $\mathcal{D}$ contains infinitely many distances, we cannot evaluate the BetaCV for each distance, one by one. Rather, we solve this optimization problem using gradient descent. In Appendix C, we prove the that BetaCV is differentiable with respect to the parameters $\alpha, \gamma, \epsilon$ and the gradient can be computed in $O(T^2 N^2)$ time, where $T$ is the number of trajectories in our dataset and $N$ is the number of elements in each trajectory (see Proposition 2).

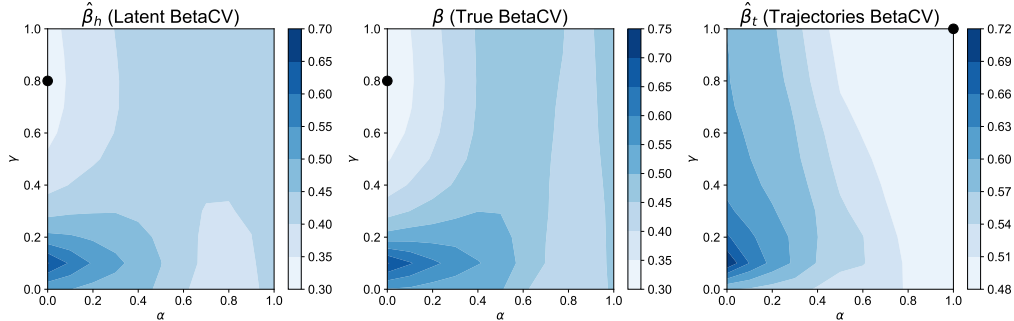

Figure 3: **Validating Latent BetaCV.** We construct a synthetic time series dataset with Gaussian noise and outliers added to the trajectories. We compute the latent betaCV for various distances (left), which closely matches the plot of the true betaCV (middle) computed based on knowledge of the seed clusters. As a control, we plot the betaCV computed based on the original trajectories (right). Black dots represent the optimal value of $\alpha$ and $\gamma$ in each plot. Lower betaCV is better.

**Batched gradient descent.** When the size of the dataset becomes modestly large, it is no longer feasible to re-compute the exact analytical gradient at each step of gradient descent. Instead, we take inspiration from negative sampling in word embeddings [28], and only sample a fixed number, $S$, of pairs of trajectories at each step of gradient descent. This reduces the runtime of each step of gradient descent to $O(SN^2)$, where $S \approx 32 - 128$ in our experiments. Instead of the full gradient descent, this effectively becomes batched gradient descent. The complete algorithm for batched Autowarp is shown in Algorithm 1 in Appendix B. Because the betaCV is not convex in terms of the parameters $\alpha, \gamma, \epsilon$, we usually repeat Algorithm 1 with multiple initializations and choose the parameters that produce the lowest betaCV.

## 4 Validation

Recall that Autowarp learns a distance from unlabeled trajectory data in two steps: first, a latent representation is learned for each trajectory; secondly, a warping distance is identified that is most similar to the learned latent representations. In this section, we empirically validate this methodology.

**Validating latent betaCV.** We generate synthetic trajectories that are copies of a seed trajectory with different kinds of noise added to each trajectory. We then measure the $\hat{\beta}_h$ for a large number of distances in $\mathcal{D}$. Because these are synthetic trajectories, we compare this to the true $\beta$ measured using the known cluster labels (each seed generates one cluster). As a control, we also consider computing the betaCV based on the Euclidean distance of the original trajectories, rather than the Euclidean distance between the latent representations. We denote this quantity as $\hat{\beta}_t$ and treat it as a control.

Fig. 3 shows the plot when the noise takes the form of adding outliers and Gaussian noise to the data. The betaCVs are plotted for distances $d$ for different values of $\alpha$ and $\gamma$ with $\epsilon = 1$. Plots for other kinds of noise are included in Appendix A (see Fig. S7). These plots suggest that $\hat{\beta}_h$ assigns each distance a betaCV that is representative of the true clustering labels. Furthermore, we find that the distances that have the lowest betaCV in each case concur with previous studies that have studied the robustness of different trajectory distances. For example, we find that DTW ($\alpha = 0.5, \gamma = 0$) is the appropriate distance metric for resampled trajectories, Euclidean ($\alpha = 1, \gamma = 0$) for Gaussian noise, and edit distance ($\alpha = 0, \gamma \approx 0.4$) for trajectories with outliers.

**Ablation and sensitivity analysis.** Next, we investigate the sensitivity of the latent betaCV calculation to the hyperparameters of the algorithm. We find that although the betaCV changes as the threshold changes, the relative ordering of different warping distances mostly remains the same. Similarly, we find that the dimension of the hidden layer in the autoencoder can vary significantly without significantly affecting qualitative results (see Fig. 4). For a variety of experiments, we find that a reasonable number of latent dimensions is $\approx L \cdot D$, where $L$ is the average trajectory length and $D$ the dimensionality. We also investigate whether both the autoencoder and the search through warping distances are necessary for effective metric learning. Our results indicate that both are

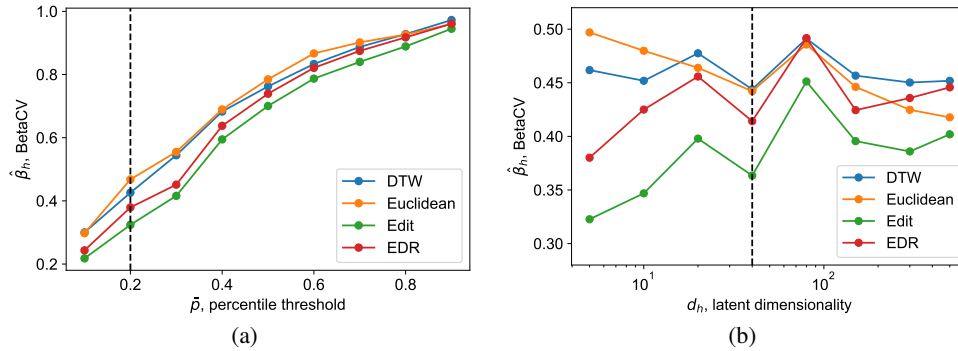

(a)                                                                (b)

Figure 4: **Sensitivity Analysis on Trajectories with Outliers.** (a) We investigate how the percentile threshold parameter affects latent betaCV. (b) We also investigate the effect of changing the latent dimensionality on the relative ordering of the distances. We find that the qualitative ranking of different distances is generally robust to the choice of these hyperparameters.

needed: using the latent representations alone results in noisy clustering, while the warping distance search cannot be applied in the original trajectory space to get meaningful results (Fig. 1).

**Downstream classification.** A key motivation of distance metric learning is the ability to perform downstream classification and clustering tasks more effectively. We validated this on a real dataset: the Libras dataset, which consists of coordinates of users performing Brazilian sign language. The $x$- and $y$-coordinates of the positions of the subjects' hands are recorded, as well as the symbol that the users are communicating, providing us labels to evaluate our distance metrics.

For this experiment, we chose a subset of 40 trajectories from 5 different categories. For a given distance function $d$, we iterated over every trajectory and computed the 7 closest trajectories to it (as there are a total of 8 trajectories from each category). We computed which fraction of the 7 shared their label with the original trajectory. A good distance should provide us with a higher fraction. We evaluated 50 distances: 42 of them were chosen randomly, 4 were well-known warping distances, and 4 were the result of performing Algorithm 1 from different initializations. We measured both the betaCV of each distance, as well as the accuracy. The results are shown in Fig. 5, which shows a clear negative correlation (rank correlation is $= 0.85$) between betaCV and label accuracy.

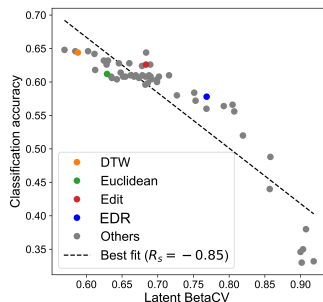

Figure 5: **Latent BetaCV and Downstream Classification.** Here, we choose 50 warping distances and plot the latent betaCV of each one on the Libras dataset, along with the average classification when each trajectory is used to classify its nearest neighbors. Results suggest that minimizing latent betaCV provides a suitable distance for downstream classification.

## 5  Autowarp Applied to Real Datasets

Many of the hand-crafted distances mentioned earlier in the manuscript were developed for and tested on particular time series datasets. We now turn to two such public datasets, and demonstrate how Autowarp can be used to *learn* an appropriate warping distance from the data. We show that the warping distance that we learn is competitive with the original hand-crafted distances.

**Taxicab Mobility Dataset.** We first turn to a dataset that consists of GPS measurements from 536 San-Francisco taxis over a 24-day period[2]. This dataset was used to test the SSPD distance metric for

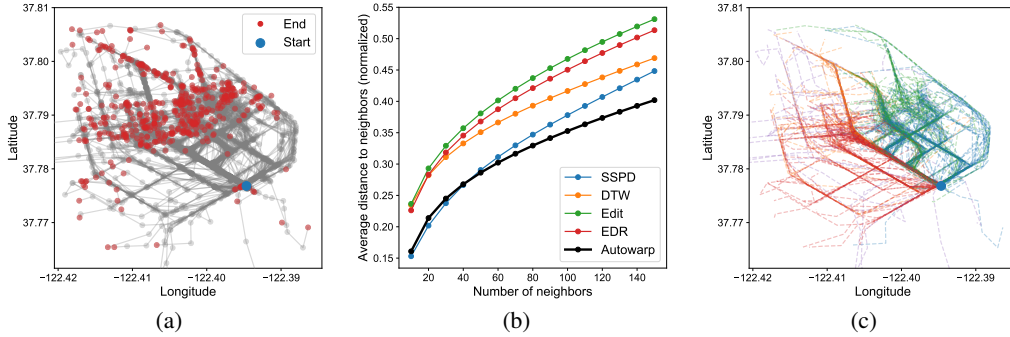

<div align="center">(a)            (b)            (c)</div>

Figure 6: **Taxicab Mobility Dataset** (a) We plot the trajectories, along with their start and end points. (b) We evaluate the average normalized distance to various numbers of neighbors for five different trajectory distances, and find that the Autowarp distance (black line) produces the most compact clusters (c) We apply spectral clustering with 5 different clusters (each color represents a different cluster) using the Autowarp learned distance.

trajectories [7]. Like the authors of [7], we preprocessed the dataset to include only those trajectories that begin when a taxicab has picked up a passenger at the the Caltrain station in San Francisco, and whose drop-off location is in downtown San Francisco. This leaves us $T = 500$ trajectories, with a median length of $N = 9$. Each trajectory is 2-dimensional, consisting of an x- and y-coordinate. The trajectories are plotted in Fig. 6(a). We used Autowarp (Algorithm 1 with hyperparameters $d_h = 10, S = 64, \bar{p} = 1/5$) to learn a warping distance from the data ($\alpha = 0.88, \gamma = 0, \epsilon = 0.33$). This distance is similar to Euclidean distance; this may be because the GPS timestamps are regularly sampled. The small value of $\epsilon$ suggests that some thresholding is needed for an optimal distance, possibly because of irregular stops or routes taken by the taxis.

The trajectories in this dataset are not labeled, so to evaluate the quality of our learned distance, we compute the average distance of each trajectory to its $k$ closest neighbors, normalized. This is analogous to how to the original authors evaluated their algorithm: the lower the normalized distance, the more "compact" the clusters. We show the result of our Fig. 6(b) for various values of $k$, showing that the learned distance is as compact as SSPD, if not more compact. We also visualize the results when our learned distance metric is used to cluster the trajectories into 5 clusters using spectral clustering in Fig. 6(c).

**Australian Sign Language (ASL) Dataset.** Next, we turn to a dataset that consists of measurements taken from a smart glove worn by a sign linguist[3]. This dataset was used to test the EDR distance metric [12]. Like the original authors, we chose a subset consisting of $T = 50$ trajectories, of median length $N = 53$. This subset included 10 different classes of signals. The measurements of the glove are 4-dimensional, including x-, y-, and z-coordinates, along with the rotation of the palm.

We used Autowarp (Algorithm 1 with hyperparameters $d_h = 20, S = 32, \bar{p} = 1/5$) to learn a warping distance from the data (learned distance: $\alpha = 0.29, \gamma = 0.22, \epsilon = 0.48$). The trajectories in this dataset are labeled, so to evaluate the quality of our learned distance, we computed the accuracy of doing nearest neighbors on the data. Most distance functions achieve a reasonably high accuracy on this task, so like the authors of [12], we added various sources of noise to the data. We evaluated the learned distance, as well as the original distance metric on the noisy datasets, and find that the learned distance is significantly more robust than EDR, particularly when multiple sources of noise are simultaneously added, denoted as "hybrid" noises in Fig. 7.

Figure 7: **ASL Dataset.** We use various distance metrics to perform nearest-neighbor classifications on the ASL dataset. The original ASL dataset is shown on the left, and various synthetic noises have been added to generate the results on the right. 'Hybrid1' is a combination of Gaussian noise and outliers, while 'Hybrid2' refers to a combination of Gaussian and sampling noise.

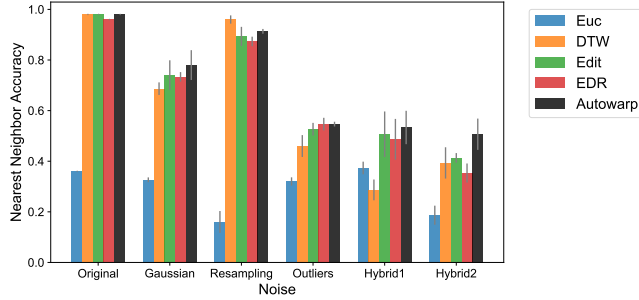

# 6 Discussion

In this paper, we propose Autowarp, a novel method to learn a similarity metric from a dataset of unlabeled trajectories. Our method learns a warping distance that is similar to latent representations that are learned for a trajectory by a sequence-to-sequence autoencoder. We show through systematic experiments that learning an appropriate warping distance can provide insight into the nature of the time series data, and can be used to cluster, query, or visualize the data effectively.

Our experiments suggest that both steps of Autowarp – first, learning latent representations using sequence-to-sequence autoencoders, and second, finding a warping distance that agrees with the latent representation – are important to learning a good similarity metric. In particular, we carried out experiments with deeper autoencoders to determine if increasing the capacity of the autoencoders would allow the autoencoder alone to learn a similarity metric. Our results, some of which are shown in Figure S5 in Appendix A, show that even deeper autoencoders are unable to learn useful similarity metrics, without the regularization afforded by restricting ourselves to a family of warping distances.

Autowarp can be implemented efficiently because we have defined a differentiable, parametrized family of warping distances over which it is possible to do batched gradient descent. Each step of batched gradient descent can be computed in time $O(SN^2)$, where $S$ is the batch size, and $N$ is the number of elements in a given trajectory. There are further possible improvements in speed, for example, by leveraging techniques similar to FastDTW [29], which can approximate any warping distance in linear time, bringing the run-time of each step of batched gradient descent to $O(SN)$.

Across different datasets and noise settings, Autowarp is able to perform as well as, and often better, than the hand-crafted similarity metric designed specifically for the dataset and noise. For example, in Figure 6, we note that the Autowarp distance performs almost as well as, and in certain settings, even better than the SSPD metric on the Taxicab Mobility Dataset, for which the SSPD metric was specifically crafted. Similarly, in Figure 7, we show that the Autowarp distance outperforms most other distances on the ASL dataset, including the EDR distance, which was validated on this dataset. These results confirm that Autowarp can learn useful distances without prior knowledge of labels or clusters within the data. Future work will extend these results to more challenge time series data, such as those with higher dimensionality or heterogeneous data.

## Acknowledgments

We are grateful to many people for providing helpful suggestions and comments in the preparation of this manuscript. Brainstorming discussions with Ali Abdalla provided the initial sparks that led to the Autowarp algorithm, and discussions with Ali Abid were instrumental in ensuring that the formulation of the algorithm was clear and rigorous. Feedback from Amirata Ghorbani, Jaime Gimenez, Ruishan Liu, and Amirali Aghazadeh was invaluable in guiding the experiments and analyses that were carried out for this paper.

## Footnotes

[1]A more general definition of warping distance replaces the summation over $c(p_{i-1}, p)$ with a general class of statistics, that may include $\max$ and $\min$ for example. For simplicity, we present the narrower definition here.

[2]Data can be downloaded from `https://crawdad.org/epfl/mobility/20090224/cab/`.

[3]Data can be downloaded from `http://kdd.ics.uci.edu/databases/auslan/auslan.data.html`.
