[Supplementary Material]

# Supplementary Materials

## A   Additional Figures

Figure S1: **Results of the Sequence-Sequence Autoencoder.** Here, we generate a synthetic dataset consisting of 5 copies of 5 trajectories. We then add sampling noise (middle column) or outliers (right column) to the trajectories. We measure the Euclidean distance between the trajectories in the original space (top row) and in the latent space learned by the autoencoder (bottom row). Results suggest that the autoencoder is partially able to recover the structure of the original trajectories.

Figure S2: **Illustration of Warping Distances.** Warping distances are defined based on an optimization over warping paths illustrated above.

Figure S3: **Illustration of Libras Dataset.** These trajectories that are used in Section 4 are shown here. Each color represents a different class (although these labels are not available to the autowarp algorithm).

Figure S4: **The Effect of Training on the Latent betaCV.** Here, we plot the latent betaCV of the distance learned by Autowarp as a function of the number of batches of examples seen during training. We find that the latent betaCV *does* decrease showing that the quality of the autoencoder does have a measurable effect on the final learned Autowarp distance.

Figure S5: **Effect of Increasing Autoencoder Complexity.** Here, we extend Fig. 1 by including a more complex, 3-layer sequence-to-sequence autoencoder to investigate whether the more powerful autoencoder is capable of learning the latent representation on its own. We do not discern any visible differences by using the more complex autoencoder (shown in the 3rd panel from the left). The complete Autowarp algorithm (shown on the right, is able to determine the similarity between related trajectories).

Figure S6: **Illustration of ASL Dataset.** These trajectories that are used in Section 5 are shown here. Each color represents a different class (although these labels are not available to the autowarp algorithm).

Figure S7: **Plots of BetaCV.** Here, we show the plots of latent betaCV (left), true betaCV (middle), and a control in the form of betaCV computed on the original trajectory (right). Each row represents a different noise source. From top to bottom: the first row is the addition outliers, the second row is the addition of Gaussian noise, the third row is sampling noise, and the fourth is a combination of Gaussian noise and outliers.

# B  Autowarp Algorithm

---

**Algorithm 1** Batched Autowarp

---

**Inputs:** Set of trajectories $\mathcal{T} = (\boldsymbol{t}^1, \dots \boldsymbol{t}^T)$, threshold percentile $p$, learning rate $r$, batch size $S$
Use a sequence-to-sequence autoencoder trained to minimize the reconstruction loss of the trajectories to learn a latent representation $h_i$ for each trajectory $\boldsymbol{t}^i$.
Compute the pairwise Euclidean distance matrix between each pair of latent representations
Compute the threshold distance $\delta$ defined as the $p^{\text{th}}$ percentile of the distribution of distances
Initialize the parameters $\alpha, \gamma, \epsilon$ (e.g. randomly between 0 and 1)
**while** parameters have not converged **do**
    Sample $S$ pairs of trajectories with distance in the latent space $< \delta$ (denote the set of pairs as $\mathcal{P}_c$), and $S$ pairs of trajectories from all possible pairs (denote the set of pairs as $\mathcal{P}_{all}$).
    Define $d$ to be the warping distance parametrized by $\alpha, \gamma, \epsilon$
    Compute the $\hat{\beta}$ as follows: $\hat{\beta} = \sum_{\boldsymbol{t}^i, \boldsymbol{t}^j \in \mathcal{P}_c} d(\boldsymbol{t}^i, \boldsymbol{t}^j) / \sum_{\boldsymbol{t}^i, \boldsymbol{t}^j \in \mathcal{P}_{all}} d(\boldsymbol{t}^i, \boldsymbol{t}^j)$
    Compute analytical gradients and update parameters:

$$\alpha \leftarrow \alpha - r \cdot d\hat{\beta}/d\alpha, \ \gamma \leftarrow \gamma - r \cdot d\hat{\beta}/d\gamma, \ \epsilon \leftarrow \epsilon - r \cdot d\hat{\beta}/d\epsilon$$

**end while**
**Return:** final betaCV $\hat{\beta}$, and the optimal parameters $\alpha, \gamma, \epsilon$

---

# C  Proofs

## C.1  Proof of Proposition 1

**Proposition 1 (Robustness of Latent BetaCV).** *Let $d$ be a trajectory distance defined over a set of trajectories $\mathcal{T}$ of cardinality $T$. Let $\beta(d)$ be the betaCV computed on the set of trajectories using the true cluster labels $\{C(i)\}$. Let $\hat{\beta}(d)$ be the betaCV computed on the set of trajectories using noisy cluster labels $\{\tilde{C}(i)\}$, which are generated by independently randomly reassigning each $C(i)$ with probability $p$. For a constant $K$ that depends on the distribution of the trajectories, the probability that the latent betaCV changes by more than $x$ beyond the expected $Kp$ is bounded by:*

$$\Pr(|\beta - \hat{\beta}| > Kp + x) \le e^{-2Tx^2/K^2} \tag{5}$$

*Proof.* The mild distributional assumption mentioned in the proposition is to ensure that no cluster of trajectories is too small or far away from the other clusters. More specifically, let $d_{\max}$ be the largest distance between any two trajectories, and let $\bar{d}$ be the average distance between all pairs of trajectories. We will define $C_1 \overset{\text{def}}{=} d_{\max}/\bar{d}$. Furthermore, let $C_2$ be defined as the number of trajectories in the largest cluster divided by the number of trajectories in the smallest cluster.

We will proceed with this proof in two steps: first, we will bound the probability that $|\hat{\beta} - E[\hat{\beta}]| > x$. Then, we will compute $|\beta - E[\hat{\beta}]|$. Then, by the triangle inequality, the desired result will follow.

Consider the effect of single trajectory $\boldsymbol{t}^i$ changing clusters from its original cluster to a new cluster. What effect does this have on the $\hat{\beta}$? At most, this will increase the distance between $\boldsymbol{t}^i$ and other trajectories with the same label by $d_{\max}$. By considering the number of such trajectories and normalizing the average distance between all pairs of trajectories, we see that the change in $\hat{\beta}$ is bounded to be less than $C_1 C_2/T$.

Since have assumed that the probability that each of the $T$ cluster labels is independently reassigned, We apply McDiarmid's inequality to bound $|\hat{\beta} - E[\hat{\beta}]|$. A straightforward application of McDiarmid's inequality shows that

$$\Pr(|\hat{\beta} - E[\hat{\beta}]| > x) \le e^{\frac{-2Tx^2}{C_1^2 C_2^2}}$$

Now, consider the difference $|\beta - E[\hat{\beta}]|$. As mentioned before, if a single trajectory $\boldsymbol{t}^i$ changes clusters from its original cluster to a new cluster, the effect is bounded by $C_1 C_2/n$. It is easy to see if subsequent trajectories change cluster assignment, the change in $\hat{\beta}$ will be of a smaller magnitude, as subsequent trajectories will face no larger of a penalty for switching cluster assignments (and in fact, may face a smaller one if they switch to a cluster that also has trajectories from the same original cluster).

Thus, we see that the $|\beta - E[\hat{\beta}]| \leq \frac{C_1 C_2}{T}(pT) = C_1 C_2 p$. Let us define $K \stackrel{\text{def}}{=} C_1 C_2$. Then applying the Triangle Inequality gives us:

$$\Pr(|\beta - \hat{\beta}| > Kp + x) \leq 4e^{-2Tx^2/K^2}$$

as desired. $\qquad\square$

### C.2 Proof of Proposition 2

**Proposition 2 (Differentiability of Latent BetaCV).** *Let $\mathcal{D}$ be a family of warping distances that share a cost function $c(\cdot)$, parametrized by $\theta$. If $c(\cdot)$ is differentiable w.r.t. $\theta$, then the latent betaCV computed on a set of trajectories is also differentiable w.r.t. $\theta$ almost everywhere.*

*Proof.* We will prove this theorem in the following way: first, we will show that any warping distance can be computed using dynamic programming using a finite sequence of only two kinds of operations: summations and minimums; then, we will show that these operations preserve differentiability except at most on a set of points of measure 0. Once, we have shown this, it is trivial to show that the latent betaCV, which is simply a summation and a quotient of such distances, is also differentiable almost everywhere.

**Lemma 1.** *Let $d \in \mathcal{D}$ be a warping distance with a particular cost function $c(\cdot)$. Then, $d(\boldsymbol{t}^A, \boldsymbol{t}^B)$, the distance between trajectories $\boldsymbol{t}^A$ and $\boldsymbol{t}^B$, can be computed recursively with dynamic programming.*

*Proof.* We prove this by construction. Let $D(i, j)$ be defined in the following recursive manner: $D(0,0) = 0$, and

$$D(i,j) = \min \begin{cases} D(i-1,j) + c(\boldsymbol{t}^A_{i-1}, \boldsymbol{t}^B_j, \boldsymbol{t}^A_n, \boldsymbol{t}^B_m) & n >= 1 \\ D(i,j-1) + c(\boldsymbol{t}^A_n, \boldsymbol{t}^B_{m-1}, \boldsymbol{t}^A_n, \boldsymbol{t}^B_m), & m >= 1 \\ D(i-1,j-1) + c(\boldsymbol{t}^A_{n-1}, \boldsymbol{t}^B_{m-1}, \boldsymbol{t}^A_n, \boldsymbol{t}^B_m) & n,m >= 1 \end{cases}$$

We will show that $D(n, m)$, where $n$ is the length of $\boldsymbol{t}^A$ and $m$ is the length of $\boldsymbol{t}^B$, is the minimal cost evaluated across all warping paths that begin at $(\boldsymbol{t}^A_0, \boldsymbol{t}^B_0)$ and end at $(\boldsymbol{t}^A_n, \boldsymbol{t}^B_m)$. By the definition of a warping distance, it follows that $d(\boldsymbol{t}^A, \boldsymbol{t}^B) = D(n, m)$.

It is obvious that if both trajectories are of zero length, then $D(0,0) = d(\boldsymbol{t}^A, \boldsymbol{t}^B) = 0$ as desired. For clarity, let us consider the base cases $n = 0, m > 0$ and $n > 0, m = 0$. For the former, clearly, there is only one warping path from $(\boldsymbol{t}^A_0, \boldsymbol{t}^B_0)$ to $(\boldsymbol{t}^A_0, \boldsymbol{t}^B_m)$. The cost evaluated over this path is:

$$0 + c((\boldsymbol{t}^A_0, \boldsymbol{t}^B_0), (\boldsymbol{t}^A_0, \boldsymbol{t}^B_1)) + \ldots + c((\boldsymbol{t}^A_0, \boldsymbol{t}^B_{m-2}), (\boldsymbol{t}^A_0, \boldsymbol{t}^B_{m-1})) + c((\boldsymbol{t}^A_0, \boldsymbol{t}^B_{m-1}), (\boldsymbol{t}^A_0, \boldsymbol{t}^B_m)))$$

$$= D(0, m-1) + c((\boldsymbol{t}^A_0, \boldsymbol{t}^B_{m-1}), (\boldsymbol{t}^A_0, \boldsymbol{t}^B_m))$$

as desired. The expression for the base cases $m = 0, n > 0$ is similarly derived.

Now, consider $m > 0, n > 0$, and let $p^* = [p_1 \ldots p_L]$ be an optimal warping path for $(A_n, B_m)$, i.e. one with a minimal cost. Note that $p_{L-1}$ must be one of

$$\{(\boldsymbol{t}^A_{n-1}, \boldsymbol{t}^B_m), (\boldsymbol{t}^A_n, \boldsymbol{t}^B_{m-1}), (\boldsymbol{t}^A_{n-1}, \boldsymbol{t}^B_{m-1})\}$$

and furthermore, we claim that it must be that an optimal path to $p_{L-1}$, denote it as $q^*$, be exactly $[p_1 \ldots p_{L-1}]$. Otherwise, we could replace $p_1 \ldots p_{L-1}$ in $p^*$ with $q^*$ and get a lower-cost optimal path, because addition is monotonically increasing. As a result, this means that we can compute the optimal warping path to $(\boldsymbol{t}^A_n, \boldsymbol{t}^B_m)$, we need to only consider the optimal paths that go through the three pairs listed above, and choose the one with the smallest total cost. This is what the recursive

equation computes, showing that $D(n, m)$ is the minimal cost evaluated across all warping paths that begin at $(t_0^A, t_0^B)$ and end at $(t_n^A, t_m^B)$. □

Now, we return to the proof of the proposition. Using Lemma 1, we see that the distance between two trajectories can be computed using a finite sequence of only two kinds of operations: summations and minimums. This also shows that the distance between two trajectories can be computed in $O(mn)$ time. We now turn to Lemma 2, which will show that these operations preserve differentiability at all except a countable number of points.

**Lemma 2.** *Let $f$ and $g$ be two continuous functions from $R^k \rightarrow R$ that are differentiable almost everywhere. Then $\min(f, g)$ and $f + g$ are also differentiable almost everywhere.*

*Proof.* Let $E_1$ be the set of points that $f$ is *not* differentiable on, and let $E_2$ be the set of points that $g$ is *not* differentiable on. Let $E$ be the set of points that $\min(f, g)$ is not differentiable on. Now, consider a point $\theta$ such that $f(\theta) \neq g(\theta)$. Without loss of generality, we can consider that $f(\theta) > g(\theta)$. Then, by continuity, $f > g$ in a neighborhood around $\theta$, and so $\min(f, g) = g$, and the derivative of $\min(f, g)$ is simply $g'$.

On the other hand, if $f(\theta) = g(\theta)$, consider the derivatives $df/d\theta_i$ and $dg/d\theta_i$. If the derivatives are all identical, then clearly the gradient of the minimum exists at $\theta$: it is simply the gradient of $f$ or $g$. If any of the derivatives, say $dg/d\theta_j$ is unequal, then the gradient of $\min(f, g)$ will not exist at $\theta$, but then without loss of generality, we can say that $df/d\theta_j > dg/d\theta_j$. In that case, there exists a neighborhood around theta in the direction of $+\theta_j$ where $f > g$ (and so the derivative of $\min(f, g)$ there is simply $g'$), and similarly, the neighborhood in the direction of $-\theta_j$ where $f < g$ (so the derivative of $\min(f, g)$ there is simply $f'$). Thus, we see that $\theta$ is an isolated point, and the set $E$ of all such $\theta$ is of measure 0.

The differentiability of $f + g$ is much easier to consider: the set of all points where $f + g$ is not differentiable is at most the union of $E_1$ and $E_2$, each are of which measure 0, so the result follows. □

It can be seen by repeatedly applying Lemma 2 that the computation of a warping distance between trajectories is differentiable. Now note that the betaCV is defined simply as the quotient of a sum of distances. We have already shown that the sum of differentiable functions is differentiable. The same can be shown for the quotient of functions, as long as the divisor is non-zero. Because the warping distance is greater than 0 as long as the trajectories are distinct, the desired result follows for any set of distinct trajectories. Finally, let us note that if there are $T$ trajectories of length $N$ in our dataset, the total runtime to compute the betaCV is $O(N^2 T^2)$.

□