[Reviews · NeurIPS 2018]

Reviewer 1



In this paper, authors proposed a metric called warping distance to measure the distance between raw sequence. BetaCV is optimized to learn the parameters in the metric and the robustness of this metric to initial guess of clustering is proven. Compared with using Euclidean distance between sequences’ latent representation, the proposed method shows some potentials to get better clustering results. My main concerns include: 1. I think authors may underestimate the power of autoencoder. The LSTM-Autoencoder used in this paper is too simple. If a VAE structure is introduced to represent latent variable h, a better reconstruction result may be available and the latent variable h may have better clustering structure. In other words, without trying more powerful autoencoder, the comparison between the proposed method and latent representation+Euclidean distance is unfair to some degree. 2. Additionally, when raw data “t” are high-dimensional observations, the distance defined in original high-dimensional space may not work well. Is it a potential risk of proposed method? 3. The scalability of the proposed method is a problem. Even if applying SGD, the complexity of the proposed method is O(N^2). When each sequence has a lot of samples (i.e., large N), which is very common in practice, the proposed method is inapplicable. 4. In the experimental section, authors seem to ignore the comparison between the proposed method with more fancy “latent representation+Euclidean distance” strategies. 5. The notations in the Proposition 1 is incomplete. The x is not well-defined. --------------------- After rebuttal ----------------------- I agree with Reviewer 2 and 3 that this work has some interesting points, e.g., learning a distance metric in an unsupervised way and unifying some distance metrics with a common parameterization. My main concern is the solidness of the experiments --- 1) I still think that VAE, especially its variant with GMM prior, should be considered as a baseline. 2) 10-dimensional data actually is not real high-dimensional data in machine learning and data science. In summary, the curse of dimensionality should be taken more seriously and more baselines should be considered.

Reviewer 2



summary: 6/10 This paper proposes a warping procedure in an unsupervised manner. Specifically, it has two separate stages. First it uses an auto encoder to learn a latent embedding for trajectories. Second, it uses a conventional metric betaCV and optimizes the average distance of trajectories within the same cluster determined by their latent representations. A few experiments were conducted on both synthetic and real data. The results can be more convincing. quality: 5/10 There are places which are a bit hand wavy. For example, in the `Warping distances` section, it is mentioned `distances are generally well-suited to trajectories, this serves to regularize the process of distance metric learning, and generally produces better distances than using the latent representations directly `. It is nice to investigate/elaborate this as one natural questions is why not simply use the Euclidian distance induced by the learned latent representation as the distance metric but still using the second stage? A proposition is provided suggesting the latent betaCV may be a reliable metric for the ground truth. But it would be nice if this proposition has any practical indications, e.g., how long the trajectory length needs to be for a set of trajectories with C clusters. As in the current work, there’s not much insight from the proposition. 3 experiments were conducted: 1 synthetic, 1 real, 1 semi-real (real data injected with noise). However, there are some questionable parts about each of the experiment. In the synthetic data, only additive noise and outliers are added; sampling rate is not mentioned but this is a main motivating point for this work. Quantitively addressing the robustness to sampling rate should be an important justification. In the real data, the metric is a bit heuristic, i.e., the closeness within same clusters varying with number of clusters. There are only 500 trajectories in total and tens of clusters. It is not convincing how strong this metric correlates with the quality of the warping procedure. Not to mention SSPD is better than Autowarp when number of neighbors is relatively small, which is a more reasonable regime to care about (e.g., <50). In the semi-real data where noise is injected. However, clearly those injected noise follow the model assumptions used in the proposed procedure, which is an unfair comparison for other procedures. This makes this ‘real data’ experiment much less convincing. clarity: 8/10 This paper is in general clearly presented. originality: 7/10 It is nice that the distance metrics are summarized with a common parametrization. The tools (auto encoders, optimization method) are conventional, but they are used in this context in an interesting way. significance: 7/10 Comparing time series is an important problem. If the proposed method is investigated in a more rigorous way, it can be more impactful. Update: The authors addressed my concerns reasonably well, though the experiments can be much stronger than presented in the paper.

Reviewer 3



## After reading the rebuttal: I maintain my initial overall score: 7. This paper proposes an unsupervised method to learn a warping distance for unlabeled time series. In contrast to the typical handcrafted distance metrics, the learned metric is expected to automatically fit the target the data and capture the complex nature of the noise in the data. Though the method is performed in an unsupervised way, it employs the autoencoder to learn the hidden representations for paired time series and calculate their euclidean distance to determine their similarity. Based on the similarity information, the distance metric is optimized by minimizing the betaCV from a family of warping distances. Strengths: 1. The method is novel in that it aims to learn a distance metric automatically in an unsupervised way. The proposed family of distances can be related to the typical handcrafted distance metrics, such as DTW, Euclidean Edit distance and so on. 2. The experiments on real datasets actually shows that the learned metric achieves competitive performance to traditional distance metric. Weaknesses: 1. On the one hand, the paper attempts to leverage the representation power of autoencoders. However, it proves in Proposition 1 that the reliability (quality) of latent representations by autoencoders (the labels) does not affect the performance significantly. It seems that the autoencoders do not play an important role in the method. I am curious about the experiments to show the effect of the quality of hidden representations by autoencoders on the performance of learned distance metric. E.g., would a poor autoencoders lead to a poor distance metric learned by autowarp?